# Metabolic Characteristics of Obese Adolescents with Different Degrees of Weight Loss After Identical Exercise Training Intervention

**DOI:** 10.3390/metabo15050313

**Published:** 2025-05-07

**Authors:** Xianyan Xie, Gaoyuan Yang, Yulin Qin, Yu Wang, Zhijun Liu, Zhuofan Zhang, Ziyan Li, Huiguo Wang, Lin Zhu

**Affiliations:** 1College of Sport and Health, Guang Zhou Sport University, Guangzhou 510500, China; xiexianyan2024@163.com (X.X.); 13682825737@163.com (G.Y.); q2320139927@163.com (Y.Q.); wangyu21077@126.com (Y.W.); 17836215062@163.com (Z.L.); zhuofanzhang2001@163.com (Z.Z.); liziyan3373972157@outlook.com (Z.L.); 2Innovative Research Center for Sports Science in the Guangdong-Hong Kong-Macao Greater Bay Area, Guangzhou 510500, China; 3Research Center for Innovative Development of Sports and Healthcare Integration, Guangzhou 510500, China

**Keywords:** metabolomics, weight-loss effect, obese adolescents, body morphology, body composition

## Abstract

**Objectives:** This study aims to elucidate the metabolic differences between obese adolescents categorized into low-weight-loss (LWL) and high-weight-loss (HWL) groups. **Methods:** The objective of this study is to investigate the metabolic characteristics of obese adolescents, with a focus on the statistically significant individual differences observed in weight loss outcomes after the same dietary and exercise training intervention. A four-week exercise and dietary intervention was administered to the participants. Obese adolescents were categorized into LWL (with a weight loss percentage of 5–10%) and HWL (with a weight loss percentage of >10%) groups on the basis of their weight loss outcomes. Post-intervention changes in body morphology and body composition between the two groups were compared using Analysis of Covariance (ANCOVA), with gender as a covariate. Additionally, metabolic changes were analyzed in depth; differential metabolites between the groups were identified through ANCOVA adjusted for gender, followed by pathway analysis. **Results:** After the four-week exercise intervention, the body morphology and composition of the obese adolescents showed significant improvements compared with those before the intervention (*p* < 0.001). For example, weight decreased from 80.65 kg to 72.35 kg, BMI decreased from 30.57 kg/m^2^ to 27.26 kg/m^2^, waist circumference decreased from 103.64 cm to 94.72 cm, and body fat percentage decreased from 32.68% to 28.54%. Prior to the exercise intervention, no significant differences in body morphology and composition were observed between the HWL and LWL groups (*p* > 0.05). After the intervention, the HWL group demonstrated significant improvements in weight, body mass index, waist circumference, body fat percentage, fat mass, fat-free mass, body water amount, and skeletal muscle mass compared with the LWL group (*p* < 0.001). After controlling for the levels of pre-intervention metabolites, 27 differential metabolites were identified between the HWL and LWL groups. These metabolites were categorized into fatty acids, amino acids, organic acids, carnitines, indoles, benzoic acids, and carbohydrates. Notably, they were significantly enriched in the eight metabolic pathways involved in amino acid metabolism, fatty acid biosynthesis, and coenzyme A biosynthesis. **Conclusions:** A four-week exercise intervention enhanced the body morphology and physical fitness of obese adolescents, although the degree of weight loss varied among individuals. Considerable weight reduction was significantly correlated with metabolites involved in lipid, amino acid, organic acid, carbohydrate, and gut microbiota metabolism and with the enrichment of pathways involved in amino acid metabolism, fatty acid biosynthesis, and coenzyme A biosynthesis. These findings indicate that intrinsic metabolic characteristics considerably influence individual responsiveness to exercise-based weight-loss interventions.

## 1. Introduction

Obesity presents an important global challenge, particularly among adolescents [1]. According to the latest statistics from the World Health Organization (WHO) in 2022, over 390 million children and adolescents aged 5–19 are affected by overweight or obesity [2]. This condition not only severely compromises the physical health of adolescents but also adversely affects their mental well-being and social adaptability [3]. Clinical studies indicate that a weight loss of 5% or more of the initial body weight can effectively mitigate the risk of obesity-related complications, including cardiovascular disease, type 2 diabetes, and metabolic syndrome [4]. Generally, achieving a 5–10% reduction in body weight is regarded as a realistic target that yields health benefits [4]. Therefore, effective weight loss strategies have become crucial for enhancing the overall health status of this population.

Various weight reduction strategies have been developed to address the issue of adolescent obesity, and examples include dietary adjustments, pharmacological treatments, lifestyle interventions, and exercise programs [5]. Among these, exercise integrated into lifestyle interventions is recognized as a cornerstone for managing adolescent obesity and supporting long-term weight control [6]. Notably, even with identical interventions, including exercise intensity and duration, remarkable individual differences in weight loss effects are observed among obese adolescents [7]. This finding suggests the presence of differential mechanisms underlying the metabolic response to exercise in individuals undergoing weight-loss interventions. Exercise induces broad metabolic enhancements by modulating the metabolic pathways involved in regulating the metabolism of the three major macronutrients: carbohydrates, lipids, and proteins [8]. In the context of carbohydrate metabolism, physical activity acutely elevates glucose uptake by skeletal muscle (mainly via GLUT4 transport mechanisms) and depletes intramuscular glycogen reserves while fostering long-term improvements in insulin sensitivity [9]. With respect to lipid metabolism, exercise promotes fat mobilization through catecholamine-induced lipolysis and markedly increases the capacity of skeletal muscle to oxidize fatty acids [10,11]. This enhanced oxidative capacity is attributable to increased mitochondrial biogenesis and the heightened activity of relevant enzymes, such as carnitine palmitoyltransferase 1, resulting in decreased fat accumulation [12]. With regard to protein metabolism, although exercise results in some degree of muscle protein breakdown, it potently stimulates muscle protein synthesis during the recovery phase, thereby facilitating the repair, remodeling, and adaptive growth of muscle tissues [13]. These comprehensive adaptive alterations in carbohydrate, lipid, and protein metabolism collectively represent the fundamental physiological mechanism by which exercise improves metabolic health and benefits weight management.

Metabolomics employs advanced analytical platforms to detect and analyze metabolites in various biological samples, such as plasma, urine, and tissues. This approach generates complex and high-dimensional datasets that facilitate the systematic characterization of metabolic processes [14]. Currently, metabolomics has emerged as a powerful tool for investigating metabolic differences between exercise-trained and untrained populations [15]. It enables the precise identification of significant metabolic disparities between these groups, elucidating distinct adaptive mechanisms related to energy metabolism, lipid utilization, and amino acid turnover [16]. Owing to long-term systematic training, exercise-trained individuals exhibit improved mitochondrial efficiency, enhanced fatty acid oxidation, and increased metabolic flexibility, which considerably enhance endurance and exercise performance [17]. Conversely, untrained individuals, lacking regular training stimuli, often exhibit an accumulation of glycolytic byproducts and oxidative stress markers, particularly after exercise and typically characterized by low metabolic efficiency and a reduced capacity for sustained physical activity [18]. Given the essential role of metabolomics in exercise physiology and its demonstrated role in revealing metabolic distinctions, the present study employs this technique to precisely analyze metabolite profiles in biological samples.

Obesity is a metabolic disease characterized by disturbances in multiple abnormal metabolic pathways [19]. These metabolic abnormalities not only affect an individual’s energy balance and lipid storage but may also lead to systemic inflammatory responses and insulin resistance, further exacerbating the development of obesity and its associated complications [20]. Given the substantial individual differences exhibited by obese adolescents during exercise-induced weight loss and the key role of metabolites in regulating energy expenditure and lipid oxidation, a comprehensive understanding of these metabolic differences is essential for developing effective weight loss strategies. Therefore, this study aims to investigate the characteristics of metabolite alterations in obese adolescents exhibiting different magnitudes of weight loss (5–10% vs. >10%) after an exercise intervention. A particular focus is placed on how these metabolic differences might elucidate the potential mechanisms underlying the observed individual variability in weight loss outcomes. We hypothesized that, compared with obese adolescents achieving 5–10% weight loss, those achieving >10% weight loss under the same exercise conditions would exhibit a stronger association between weight loss response and specific alterations in their metabolomic profiles.

## 2. Materials and Methods

### 2.1. Selection of Subjects

A total of 103 subjects at a weight loss boot camp were recruited for this study. The inclusion criteria were (i) age of 9–17 years and (ii) a body mass index (BMI) ≥the 95th percentile of the WHO growth curve [2]. The exclusion criteria included (i) weight fluctuation exceeding 5% over the past three months, (ii) use of medications that influence glucose or lipid metabolism (such as insulin sensitizers, hormonal drugs, and orlistat) within the last six months, and (iii) the presence of other medical conditions (including diabetes mellitus and severe cardiovascular disease) that could affect the study results.

### 2.2. Grouping of the Study Subjects

The selection of the 10% weight loss threshold was based on clinical guidelines [21], which highlight the clinical significance of achieving ≥5–10% weight loss for substantial health benefits. This threshold allowed us to differentiate between levels of clinically relevant weight loss achieved during the intervention. At the conclusion of the exercise intervention, the subjects were categorized into two groups on the basis of their percentage of weight loss [21]: low weight loss (LWL, defined as a percentage of weight loss between 5% and 10%) and high weight loss (HWL, defined as a percentage of weight loss greater than 10%).

The percentage of weight loss was calculated using the formula (pre-intervention weight − post-intervention weight)/pre-intervention weight × 100%.

### 2.3. Exercise Intervention Protocol

The intervention protocol employed in this study was approved by the Ethics Committee of Guangzhou Sports Institute (No. 2018LCLL-008). Prior to the intervention, all subjects and their parents were thoroughly informed about the benefits and potential risks associated with this study, and written informed consent was obtained.

#### 2.3.1. Exercise Intervention

In accordance with the principles of individualization, progressivity, engagement, and safety, an exercise program was developed for this study. The interventions were scheduled daily from 9:30 am to 11:30 am and from 3:30 pm to 5:30 pm, with a total of 240 min per day for a duration of four weeks. All training sessions were conducted under the supervision of an experienced physical trainer. The specific training protocol was implemented as follows:

Frequency: The participants trained twice daily, seven days per week, for four consecutive weeks.

Intensity: Training intensity was predominantly moderate, and the participants maintained a target heart rate between 50% and 80% of their HRmax.

Interval: Each session comprised a 30 min warm-up, an 80 min formal training session, and a 10 min relaxation period. The main training phase employed a 1:2 work-to-recovery ratio structured as

−Work Interval: 2 min at 70–80% HRmax (moderate-to-high intensity).−Recovery Interval: 4 min at 50–70% HRmax (low-to-moderate intensity aerobic).

Type: The protocol consisted of moderate-intensity aerobic exercise interspersed with short bursts of high-intensity activity.

#### 2.3.2. Dietary Control

On the basis of the weekly resting metabolic rate of the subjects and with reference to the Chinese Food Composition Table compiled by the Chinese Centre for Disease Control and Prevention, nutritional experts tailored the dietary intake of each subject. A dietary regimen was designed to include various food types, with a scientifically allocated distribution of energy intake per day (30% for breakfast, 40% for lunch, and 30% for dinner). The subjects were instructed to adhere to a regular eating schedule, with breakfast from 8:00 a.m. to 8:30 a.m., lunch from 11:30 a.m. to 12:00 noon, and dinner from 5:30 p.m. to 6:00 p.m.

### 2.4. Clinical Indicator Testing

#### 2.4.1. Acquisition of Body Morphological Indicators

The height and weight of the subjects were respectively measured using a height meter (Suheng, Shanghai, China, SZ200) and a weight meter (Huibao, Guangdong, China, EB921). Chest, waist, and hip circumferences were also measured using a nonretractable circumference scale (Icomon, Guangdong, China, ER301). The measurements of height were recorded in centimeters (cm), with an accuracy of 0.1 cm, and all measurements of height and circumferences were recorded in centimeters (cm), with an accuracy of 0.1 cm, while weight was measured in kilograms (kg) with an accuracy of 0.01 kg.

#### 2.4.2. Acquisition of Fitness Level

The Inbody370 body composition analyzer (Seoul, Korea) was employed to evaluate the fitness level of the subjects. The participants stood in a relaxed position, ensuring that their heels were aligned with the foot electrodes, while in a fasting state during the morning. They held the handles with their arms extended and thumbs placed on the oval electrodes. The bioelectrical impedance measurements were performed at frequencies of 5, 50, and 250 kHz. The parameters obtained comprised body fat percentage, body fat mass (BMI), free fat weight, body water, and skeletal muscle mass.

#### 2.4.3. Statistical Analysis of Clinical Indicators

Statistical analysis was conducted using SPSS 26.0. Categorical count data were analyzed using the Chi-square test. For data that followed a normal distribution, paired samples t-test and independent samples t-test were employed, with the results presented as mean ± standard deviation (mean ± SD). Conversely, for data that did not adhere to a normal distribution, the Mann–Whitney U-test and the Wilcoxon signed-rank test were utilized, with the results expressed as median (first quartile, third quartile) in the format [median (Q1, Q3)]. Furthermore, to account for the potential influence of sex on the response to the intervention, we employed analysis of covariance (ANCOVA) to compare differences between groups in terms of key continuous outcomes post-intervention. We included sex as a key covariate to adjust for its potential effect. Effect sizes were calculated for each outcome using Cohen’s d. For within-group comparisons, Cohen’s d = (pre-intervention mean − post-intervention mean)/combined standard deviation. Dor between-group comparisons, Cohen’s d = (HWL group mean − LWL group mean)/combined standard deviation. Effect sizes were interpreted according to the following thresholds: very small (<0.2), small (>0.2 but <0.5), medium (>0.5 but <0.8), and large (>0.8).

### 2.5. Metabolomic Analysis

#### 2.5.1. Sample Preparation

Venous blood was collected from the anterior region of the elbow of the subjects in a fasting state during the morning one day before and one day after the intervention. Heparin was used as an anticoagulant. The blood samples were allowed to stand for 30 min and centrifuged at a low temperature of 4 °C at 3000 rpm for 15 min, followed by the extraction of the upper plasma layer. Plasma samples stored at −80 °C were thawed in an ice bath. A volume of 20 μL of the thawed blood samples was added to a 96-well plate and processed using an Eppendorf epMotion workstation (Eppendorf Inc., Hamburg, Germany). Subsequently, 120 μL of a precooled methanol solution containing the internal standard was added to each well and vortexed vigorously for 5 min to ensure thorough mixing. The plate was then centrifuged for 30 min at 4000× *g* and 4 °C. After centrifugation, 20 μL of a freshly prepared derivatization reagent was added to each well within the workstation. The plate was sealed and incubated at 30 °C for 60 min to facilitate the derivatization reaction. Upon completion of the reaction, the samples were diluted in an ice bath by adding 330 μL of a 50% methanol solution. The samples were then centrifuged again at 4000× *g* for 30 min at 4 °C. A volume of 135 μL of the supernatant was aspirated and transferred to a new 96-well plate, to which 10 μL of the internal standard was added to each well. Moreover, a gradient dilution of the derivatized standard stock solution was introduced to the left well. Then, the plate was sealed and prepared for liquid chromatography–mass spectrometry (LC-MS) analysis.

#### 2.5.2. Ultra-Performance Liquid Chromatography–Tandem Mass Spectrometry Analysis of Plasma Components

An ultra-performance liquid chromatography–tandem mass spectrometry instrument (ACQUITY UPLC-Xevo TQ-S, Waters Corp., Milford, MA, USA) was employed to detect 324 plasma metabolites in the plasma samples. For the instrumental setup, an ACQUITY UPLC BEH C18 1.7 μm VanGuard pre-column (dimension: 2.1 × 5 mm) and an ACQUITY UPLC BEH C18 1.7 μm analytical column (dimension: 2.1 × 100 mm) were utilized for sample separation. The column temperature was maintained at 40 °C, and the sample manager temperature was set to 10 °C. Sample separation was conducted on the ACQUITY UPLC BEH C18 1.7 μm analytical column. The mobile phase comprised water with 0.1% formic acid (designated as A) and a mixture of acetonitrile and isopropanol at a 70:30 ratio (designated as B). Gradient elution was subsequently performed in accordance with a predetermined elution program to ensure the effective separation of each metabolite. The flow rate was 0.4 mL/min, and the injection volume was 5.0 μL. For mass spectrometry detection, the capillary voltages were set to 1.5 kV (positive ion mode) and 2.0 kV (negative ion mode) depending on the nature of the substances being measured. The temperature of the ion source was maintained at 150 °C, and the desolvation temperature was set to 550 °C. The desolvation gas flow rate was 1000 L/h.

#### 2.5.3. Differential Metabolite Screening

Statistical analyses were conducted using the iMAP (v1.0, Metabo-Profile, Shanghai, China) platform. To identify differential metabolites between the two groups post-intervention while controlling for sex as a potential confounding factor, an ANCOVA model was employed, incorporating sex as a covariate. Based on the results from this model, the criteria for significance were set at *p* < 0.05 and |log2(Fold Change)| ≥ 0 [22]. The differentially expressed metabolites were visualized using a volcano plot, and the pre-intervention levels of metabolites with significant differences between groups after covariate adjustment were further analyzed. ANCOVA, with sex included as a covariate, was again used to assess whether or not the observed post-intervention differences already existed before the intervention began (to rule out the influence of baseline differences).

#### 2.5.4. Discriminatory Efficacy Test for Differential Metabolites

The differential metabolites were analyzed using the receiver operating characteristic (ROC) curve to assess their potential application value. A differential metabolite was deemed a potential marker for varying degrees of weight loss in obese adolescents when the area under the ROC curve (AUC) exceeded 0.5.

#### 2.5.5. Metabolic Pathway Analysis

Pathway enrichment analyses of the differential metabolites were conducted using the Kyoto Encyclopedia of Genes and Genomes (KEGG). A small *p*-value meant high enrichment of differential metabolites within the pathway, and a large impact value signified a great influence on the pathway.

## 3. Results

### 3.1. Effects of Exercise Intervention on Body Morphology and Fitness Level

Among the 103 obese adolescents, 5 withdrew midway, resulting in a final sample of 98 participants for statistical analysis. The overall percentage of weight loss for these 98 participants was 9.01% ± 3.46%.

As shown in Table 1, after the four-week exercise intervention, significant reductions in body weight, BMI, chest circumference, waist circumference, hip circumference, waist-to-hip ratio, waist-to-height ratio, body fat percentage, body fat mass, free fat weight, body water, and skeletal muscle mass were observed compared with the pre-intervention measurements (*p* < 0.001). This result indicates that the exercise intervention had a remarkable effect on improving body morphology and optimizing fitness level.

### 3.2. Changes in Body Morphology and Fitness Level Between the HWL and LWL Groups

A total of 98 subjects were categorized into two groups, namely, LWL and HWL, on the basis of their percentage of weight loss. Among the subjects, 46 participants (47% of the sample) with a percentage of weight loss between 5% and 10%, and 52 participants (53% of the sample) with a percentage of weight loss of 10% or greater.

A comparative analysis of body morphology and fitness level was conducted prior to the exercise intervention to eliminate any initial differences between the two groups. The results in Table 2 show no significant differences in body morphology and fitness level indicators between the LWL and HWL groups before the exercise intervention (*p* > 0.05). This finding suggests that the two groups of subjects had comparable body morphology and fitness level at the onset of the study.

As presented in Table 3, after a four-week exercise intervention, all body morphology and fitness level indices decreased in both groups. A comparison of the magnitude of these decrements between the two groups revealed that the HWL group experienced a larger reduction in body weight, BMI, waist circumference, hip circumference, body fat percentage, body fat mass, free fat weight, body water, and skeletal muscle mass compared with the LWL group, with the differences being extremely significant (*p* < 0.001). Moreover, the reduction in the waist-to-height ratio in the HWL group was significantly larger than that in the LWL group (*p* < 0.01). However, the changes observed in chest circumference and waist-to-hip ratio between the two groups had no statistically significant differences (*p* > 0.05).

### 3.3. Identification of Differential Metabolites Between HWL and LWL Groups Before and After Intervention

Targeted metabolomic analysis successfully quantified 204 metabolites in plasma samples from the study participants. This study aimed to characterize the metabolic differences between obese adolescents with high weight loss (HWL) and low weight loss (LWL) after an identical exercise intervention, while adjusting for baseline metabolic variations. To this end, we employed ANCOVA, with sex included as a covariate, to compare plasma metabolite levels between the HWL and LWL groups pre- and post-intervention.

As shown in Table 4, baseline analysis identified 32 metabolites with significantly different levels (*p* < 0.05) between the HWL and LWL groups, confirming initial metabolic variations between the groups. Following the four-week identical exercise intervention, 32 metabolites exhibited significantly different levels (*p* < 0.05) between the HWL and LWL groups (Table 5).

### 3.4. Differential Metabolites Between the LWL and HWL Groups After Accounting for Baseline Differences

To identify the metabolites uniquely differentiating the HWL and LWL groups post-intervention after accounting for baseline differences, we compared the sets of significantly altered metabolites identified pre- and post-intervention using a Venn diagram. Venn diagram analysis (Figure 1) revealed 27 metabolites that were significantly different (*p* < 0.05) between the HWL and LWL groups post-intervention, after accounting for baseline differences. Therefore, these 27 metabolites represent a distinct post-intervention metabolic signature associated with different magnitudes of weight loss, independent of baseline metabolic status. Meanwhile, six metabolites, namely, hydroxyphenyllactic acid, maltotriose, homovanillic acid, undecylenic acid, arachidonic acid, and DPAn-6, already exhibited significant differences at baseline.

Subsequently, the aforementioned differential metabolites were analyzed using the ROC curve method to further evaluate and compare the clinical applicability of discriminatory experiments. An AUC greater than 0.5 is indicative of diagnostic value. As presented in Table 6, the AUC for the 27 differential metabolites all exceeded 0.5, suggesting their potential use as biomarkers for distinguishing varying degrees of weight loss in obese adolescents.

### 3.5. Class Distribution of the Differential Metabolites

The type and distribution of the 27 differential metabolites and their variations were analyzed and visualized using Z-score diagrams. As illustrated in Figure 2, the 27 differential metabolites could be categorized into seven groups: fatty acids, amino acids, organic acids, carnitines, indoles, benzoic acids, and carbohydrates.

### 3.6. Pathway Analysis for the Differential Metabolites

Pathway enrichment analyses were conducted on the differential metabolites, resulting in the creation of a bubble diagram. As depicted in Figure 3, a total of 27 differential metabolites were enriched across 28 metabolic pathways. Using a threshold of *p* < 0.05 and impact values as screening criteria, we identified eight significantly enriched metabolic pathways: ① valine, leucine, and isoleucine biosynthesis; ② alanine, aspartate, and glutamate metabolism; ③ glycine, serine, and threonine metabolism; ④ arginine biosynthesis; ⑤ cysteine and methionine metabolism; ⑥ valine, leucine, and isoleucine degradation; ⑦ pantothenate and coenzyme A (CoA) biosynthesis; and ⑧ biosynthesis of unsaturated fatty acids. These pathways play critical roles in the different weight loss effects of exercise interventions in obese adolescents.

## 4. Discussion

The significant health burden imposed by adolescent obesity underscores the urgent need for effective management strategies [23]. Exercise intervention, recognized as a nonpharmacological, nonsurgical approach to weight reduction, is highly regarded for its effectiveness and sustainability [24]. However, previous studies have revealed substantial individual differences in weight loss among obese adolescents with identical basal body weights who underwent the same exercise intervention [25]. To elucidate the metabolic basis for this variability, the present study investigated the differences in the plasma metabolic profiles of obese adolescents exhibiting varying levels of weight loss, all subjected to the same exercise intervention combined with dietary control on the basis of resting metabolic rate design.

Prior to analyzing the metabolites underlying these variations, our initial observation was that the four-week intervention led to overall improvements in the participants’ physical status. We found that body morphology indicators, such as weight, BMI, chest circumference, waist circumference, hip circumference, waist-to-hip ratio, and waist-to-height ratio, and body composition indicators, including body fat percentage, body fat mass, free fat weight, body water, and skeletal muscle mass, were considerably reduced after a four-week exercise intervention. However, despite these positive outcomes, a statistically significant decrease in skeletal muscle mass was observed (Table 1). This phenomenon is not uncommon in weight loss interventions, particularly when a significant energy deficit is present [26]. In our study, it occurred despite the implementation of an exercise training intervention designed to preserve muscle mass, possibly because of a combination of energy restriction [26] and suboptimal protein intake [27] or the failure of the exercise protocol [28] to fully counteract catabolism. The potential metabolic side effects associated with skeletal muscle mass reduction should be acknowledged. Skeletal muscle constitutes one of the largest metabolically active tissues in the human body, and a decline in its mass directly leads to a reduction in basal metabolic rate [29]. This decline not only potentially diminishes the total energy expenditure advantage gained from weight loss but may also complicate long-term weight maintenance, increasing the risk of weight regain when normal diet is resumed or physical activity decreases [30]. Therefore, future weight loss intervention protocols should place greater emphasis on optimizing strategies to efficiently preserve or increase skeletal muscle mass [31]. These protocols should ensure adequate protein intake and integrate effective resistance training programs [31]. In research and clinical settings, preserving muscle mass should be regarded as equally important as weight loss in the design and evaluation of intervention strategies.

Further analysis of the impact of the exercise intervention on obese adolescents revealed significant individual differences in the magnitude of weight loss, which ranged from 5% to 18%. This finding highlights the notable heterogeneity in adolescents’ weight-loss response to exercise [32]. Although the weight loss rate of some participants exceeded the long-term health goal recommended by the ACSM (approximately 1% body weight per week), the purpose of this study was not to promote or evaluate weight loss rates exceeding these guidelines. Instead, the core objective was to objectively observe and analyze naturally occurring individual differences in weight loss rates within a realistic short-term intervention context.

The observation in this study, that is, approximately half of the participants achieved weight loss rates consistent with or below ACSM recommendations, while the other half exhibited markedly faster rates, strongly suggests that intrinsic physiological factors, such as genetics and hormones [33], alongside broad complexities inherent in body-weight-regulation models and the pathogenesis of obesity [34,35], play a crucial role alongside exercise in determining ultimate individual weight loss outcomes. However, under free-living conditions, an exercise-induced increase in energy expenditure is often accompanied by a compensatory increase in energy intake, which is a critical factor influenced by complex regulatory feedback mechanisms and individual predisposition and ultimately affects the long-term effectiveness of exercise interventions [35,36]. Individual differences in appetite signals, the relationship of which to actual intake can be intricate and not always explained by specific known hormonal pathways [37], satiety responses, and behavioral choices in response to the food environment can considerably affect the extent of this compensatory response [34,38]. This variability potentially confounds the heterogeneity observed in weight loss outcomes. Recognizing this crucial interplay, we determined the potential confounding effect of compensatory eating. A key feature of this study’s design is the strict control of the participants’ daily energy intake by dynamically adjusting energy prescriptions based on weekly individual RMR measurements. Standard food composition tables for precise meal preparation were used, and strict measures for mealtimes and food types were implemented. Thus, we aimed to temporarily “isolate” or minimize the direct impact of behavioral compensatory intake as a variable to focus on observing intrinsic physiological and metabolic adaptive differences among individuals undergoing the same exercise intervention. To elucidate the influence of intrinsic biological factors on the individualized differences in weight loss outcomes, we utilized metabolomics to compare the metabolic profiles of individuals achieving HWL (>10%) versus those with LWL (5–10%) who underwent the same exercise intervention, aiming to identify key potential metabolic mechanisms.

Sex is a key biological factor influencing metabolic profile. Differences in sex hormone levels and associated genetic backgrounds result in inherent distinctions between males and females in core metabolic pathways, including lipid metabolism, energy homeostasis, and amino acid profiles [39,40]. Given the considerable disparity in sex composition within our study cohort, sex is regarded as a potential confounding factor that could obscure the accurate identification and interpretation of unique metabolic features associated with varying degrees of weight loss. To control for this potential bias, we employed ANCOVA in our primary statistical analyses to assess the impact of weight loss magnitude on metabolites, incorporating sex as a covariate. This approach aimed to statistically adjust for the variability introduced by sex and accurately identify independent metabolic effects attributable to different degrees of weight loss. However, ANCOVA, as a linear model adjustment method, may not fully capture all complex, nonlinear sex-related biological differences and might mask potential sex-specific response patterns to weight-loss intervention [41]. Therefore, future studies incorporating sex-stratified analyses or direct interaction testing would aid in elucidating sex-specific metabolic regulatory mechanisms underlying variations in weight-loss outcomes.

Furthermore, in determining why identical exercise interventions lead to different degrees of weight loss and corresponding metabolic effects, baseline metabolic status emerges as a critical factor that warrants careful consideration. Following the four-week exercise intervention, our comparison of metabolic profiles between groups with differing weight loss outcomes identified 33 metabolites exhibiting significant differences. Notably, the baseline levels of six of these metabolites, namely, hydroxyphenyllactic acid, maltotriose, homovanillic acid, undecylenic acid, arachidonic acid, and DPAn-6, showed significant differences between the groups before the intervention commenced. This finding indicates that, for these six metabolites, the post-intervention differences observed in association with weight loss magnitude cannot be entirely attributed to the direct physiological effects induced solely by varying levels of weight loss. These differences are likely substantially influenced or confounded by the inherent and preexisting metabolic characteristics of the individuals. To accurately identify biomarkers that genuinely reflect the metabolic changes specific to the process or outcome of differential weight loss, we excluded the six metabolites, which had significant baseline inter-group differences from subsequent analyses, rather than representing legacy effects from the baseline state.

After excluding metabolites confounded by baseline differences, we proceeded to analyze the 27 metabolites that remained significantly different between the HWL and LWL groups. These metabolites spanned seven classes: fatty acids, amino acids, organic acids, carnitines, indoles, benzoic acids, and carbohydrates.

The HWL group exhibited significantly elevated levels of eight fatty-acid-related metabolites: 2-hydroxy-3-methylbutyric acid, 3-hydroxyisovaleric acid, adrenic acid, docosahexaenoic acid (DHA), dihomo-gamma-linolenic acid, docosapentaenoic acid (DPA), myristic acid, and oleic acid. This result suggests a pronounced activation of adipose tissue lipolysis and fatty acid mobilization in response to the exercise intervention in HWL individuals. Specifically, the elevation of multiple polyunsaturated fatty acids (PUFAs), such as DHA, DPA, adrenic acid, and dihomo-gamma-linolenic acid, indicates effective PUFA-mediated signaling pathway modulation during exercise adaptation in the HWL group. This elevation could manifest as favorable inflammatory state regulation or optimized metabolic signaling, thereby supporting the weight loss process [42]. Alternatively, it might reflect considerable cellular membrane lipid remodeling or alterations in tissue-specific PUFA utilization patterns [43], potentially enhancing energy expenditure efficiency or fatty acid oxidation capacity [44]. Moreover, the increased levels of 2-hydroxy-3-methylbutyric acid, which are the byproducts of branched-chain amino acid (BCAA) metabolism, suggest that the HWL group experienced greater overall metabolic stress or significantly enhanced BCAA catabolism pathways as a supplementary energy source [45]. This augmented catabolism could directly contribute to a large energy deficit, thus supporting their more substantial weight loss.

In the HWL group, pipecolic acid levels were considerably elevated, whereas the levels of seven amino-acid-related metabolites were greatly reduced: alanine, aspartic acid, cystine, serine, ornithine, phenylalanine, and threonine. This profile suggests potentially more active amino acid turnover and utilization under exercise stress in the HWL group. A reduction in multiple amino acids, particularly glucogenic amino acids such as alanine, aspartic acid, serine, and threonine, may reflect enhanced gluconeogenesis in the HWL group and met high energy demands during exercise and recovery [46]. This finding indicates that HWL individuals more effectively mobilize amino acids as energy substrates to support their greater energy expenditure and weight loss. Furthermore, the reduced ornithine level, which is a key intermediate in the urea cycle, may contribute to the alteration of the overall protein turnover or specific metabolic pathway shifts [47]. Meanwhile, pipecolic acid is an intermediate in lysine catabolism, and its presence indicates a bottleneck in downstream steps in the HWL group [48].

The HWL group displayed significantly lower Pyruvic acid levels, whereas the levels of glucaric acid, 3-methyl-2-oxopentanoic acid, 2-hydroxybutyric acid, alpha-ketoisovaleric acid, and ketoleucine were significantly elevated. This result suggests a combination of efficient core energy pathway utilization, enhanced mobilization of alternative fuels, and accompanying metabolic stress adaptation in the HWL group following the intervention. Pyruvic acid, which is a key end-product of glycolysis and a hub connecting carbohydrate, fat, and amino acid metabolism, was relatively lower in the HWL group, strongly indicating higher pyruvate metabolic flux [49]. This finding could reflect several scenarios: (1) more efficient entry of pyruvate into the tricarboxylic acid (TCA) cycle for oxidative phosphorylation to meet higher energy demands; (2) greater utilization of pyruvate for gluconeogenesis to maintain glucose homeostasis during exercise and recovery; and (3) a combination of both. Regardless of the specific route, low pyruvate levels are generally considered a marker of increased energy demand and improved metabolic efficiency, consistent with the superior weight loss outcome in the HWL group. The elevated levels of 3-methyl-2-oxopentanoic acid, alpha-ketoisovaleric acid, and ketoleucine, which are the key intermediates of BCAA catabolism [50], further corroborate the conclusions from the amino acid metabolism analysis, that is, the HWL group likely experienced enhanced BCAA breakdown, which not only provided additional carbon skeletons for the TCA cycle (via acetyl-CoA or succinyl-CoA generation) but also reflected specific metabolic signaling under conditions with high energy demands [51]. Furthermore, elevated 2-hydroxybutyrate, which often indicates cellular stress or increased metabolic flux after exercise or in chronic metabolic disorders, was observed in the HWL group [52]. This finding suggests that these individuals experienced higher exercise-induced metabolic pressure, which potentially activated the adaptive feedback loops involving BCAA metabolism and skeletal muscle oxidative capacity [52]. An increase in the level of glucaric acid was observed, which is primarily produced via the glucuronic acid pathway and contributes to detoxification by inhibiting β-glucuronidase and through uncoupling [53,54]. This metabolite could be linked to the increased release of potential toxic substances accompanying fast lipolysis.

The HWL group showed significantly lower hippuric acid levels, whereas the levels of gallic acid, acetylcarnitine, o-adipoylcarnitine, and indole-3-carboxylic acid were considerably elevated. This result suggests that, in the HWL individuals responding to exercise, a distinct interaction pattern conducive to weight loss existed between the gut microecology and host metabolism. Hippuric acid results from the hepatic conjugation of benzoic acid with glycine, and benzoic acid can be produced by gut microbiota metabolizing dietary polyphenols (such as quinic acid) [55]. Therefore, the lower hippuric acid level in the HWL group could reflect several potential mechanisms: (1) altered capacity or pattern of the gut microbiota metabolizing specific dietary polyphenol precursors; (2) changes in host glycine availability (potentially diverted to other metabolic pathways); and (3) adjustments in hepatic conjugation capacity. Given the pivotal role of gut microbiota in the polyphenol metabolism, this finding suggests the specific functional remodeling of the gut microecology in the HWL group post-intervention. This process might be linked to effective energy metabolism regulation [56]. Gallic acid, which is a phenolic acid found naturally in various plants and produced through the gut microbial hydrolysis of plant tannins and other polyphenols [57,58], was elevated in the HWL group. This elevation implies the increased capacity of their gut microbiota to degrade specific polyphenols or increased intake of precursor-rich food during the intervention (although dietary control was attempted, complete consistency was challenging). Gallic acid possesses antioxidant and anti-inflammatory bioactivities [58], and an increase its level could have contributed to the enhancement of the overall metabolic environment in the HWL group, potentially mitigating obesity-related low-grade inflammation and thus indirectly promoting weight loss. The elevated levels of acetylcarnitine and O-adipoylcarnitine are indicative of increased level of active fatty acid oxidation. Acetylcarnitine serves as a key metabolic intermediate, buffering mitochondrial acetyl-CoA, which can originate from substrates, including short-chain fatty acids. Its elevation is typically associated with increased energy demand and accelerated fatty acid β-oxidation rates, reflecting the status of fatty acid–derived energy production [59]. The presence of O-adipoylcarnitine, which is a medium- or long-chain dicarboxylic acylcarnitine, likely reflects enhanced fatty acid ω-oxidation, which is often activated as a compensatory pathway when β-oxidation is overloaded or impaired [60]. This finding provides further evidence of large-scale fat mobilization and catabolism in the HWL group. Although carnitine metabolism primarily occurs within a host, the gut microbiota can influence the host’s carnitine pool and metabolism [61]; however, in this context, these carnitines directly reflect the enhanced energy expenditure and fat utilization efficiency in the HWL group. Indole-3-carboxylic acid is a metabolite produced from tryptophan through specific gut microbial metabolic pathways (indole pathway) [62]. Indoles and their derivatives serve as important signaling molecules, modulating host gut barrier function, immune responses, and metabolic homeostasis, often by activating the aryl hydrocarbon receptor [63]. The elevated level of indole-3-carboxylic acid in the HWL group suggests that their gut microbiota possess a greater capacity to produce potentially beneficial metabolites. This feature contributes to maintaining gut health, reducing systemic inflammation, and exerting positive effects on host energy metabolism, thereby leading to considerable weight reduction.

To further understand the metabolic differences between the HWL and LWL groups post-intervention (after accounting for baseline effects) from a systems perspective, we performed pathway enrichment analysis on the 27 differentially abundant metabolites. This analysis identified eight significantly enriched metabolic pathways, which were primarily concentrated in key areas, such as amino acid metabolism, fatty acid biosynthesis, and CoA biosynthesis.

Amino acid metabolism pathways were the most prominently enriched category, with six related pathways identified. The valine, leucine, and isoleucine biosynthesis pathway emerged as the most statistically significant in our analysis. However, the elevated levels of several BCAA catabolites (e.g., 3-methyl-2-oxopentanoic acid, alpha-ketoisovaleric acid, ketoleucine, 2-hydroxybutyric acid, and 3-hydroxyisovaleric acid) in the HWL group strongly suggest enhanced BCAA degradation rather than synthesis. This finding aligns with previous discussions indicating that the HWL group likely utilized BCAAs as energy sources or as precursors for the synthesis of other essential molecules to meet high energy demands [64]. The enrichment of the valine, leucine, and isoleucine degradation pathway further corroborates this interpretation. Additionally, the enrichment of four other pathways, namely, alanine, aspartate, and glutamate metabolism; glycine, serine, and threonine metabolism; arginine biosynthesis; and ysteine and methionine metabolism, aligns with our observations of reduced levels of several amino acids (alanine, aspartic acid, cystine, serine, threonine, and phenylalanine) and the decreased level of ornithine (involved in arginine metabolism) in the HWL group. Collectively, these findings depict dynamic alterations in the amino acid pool, potentially reflecting enhanced amino acid oxidation for energy supply or the increased utilization of amino acid carbon skeletons for pathways, including gluconeogenesis, to support energy requirements and metabolic adaptation [65].

The enrichment of the unsaturated fatty acid pathway corresponded to the elevated levels of specific unsaturated fatty acids (e.g., DHA, adrenic acid, and dihomo-gamma-linolenic acid) in the HWL group. This result suggests that the HWL group exhibited a more active response in modulating lipid profiles related to cell membrane fluidity, signal transduction, and inflammatory responses following the exercise intervention [66]. These properties could be crucial for adapting to physiological changes induced by exercise.

Furthermore, pantothenate (vitamin B5) is an essential precursor for the synthesis of CoA, which plays a central role in energy metabolism and serves as a necessary cofactor in numerous key processes, including fatty acid β-oxidation, the entry of pyruvate into the TCA cycle (via acetyl-CoA), and BCAA catabolism [67]. The enrichment of the pantothenate and CoA biosynthesis pathway, coupled with the elevated acetylcarnitine and O-adipoylcarnitine levels (reflecting active fatty acid and potentially amino acid oxidation) in the HWL group strongly implies increased overall metabolic flux, particularly in central energy-producing pathways. An enhanced capacity for CoA biosynthesis may fundamentally underpin more efficient energy expenditure and substrate utilization in this group, contributing to increased weight loss.

Synthesizing these findings, this study successfully employed a metabolomics approach within a controlled intervention framework to unveil the distinct metabolic signatures associated with high versus low weight loss responses in obese adolescents subjected to identical exercise and diet protocols. The high-weight-loss achievers demonstrated a metabolic phenotype characterized by enhanced fatty acid mobilization and oxidation, dynamic amino acid turnover, and indications of potentially favorable gut microbiota–host interactions, which are all seemingly underpinned by more robust central energy pathway activity. These multifaceted metabolic distinctions advance beyond recognizing well-known heterogeneity in exercise-induced weight loss, offering insights into the underlying intrinsic physiological determinants driving these differences when behavioral factors, such as compensatory eating, are minimized. The identified metabolites and pathways represent concrete biological determinants that help explain differences in response among individuals subjected to the same energy deficit and exercise stimulus. These insights hold considerable promise in informing the development of future personalized weight management strategies. These strategies may facilitate the prediction of weight loss responsiveness or guide the tailoring of interventions, such as nutritional support or specific exercise modalities, for individual metabolic profiles, ultimately resulting in effective and sustainable outcomes in the ongoing challenge of combating adolescent obesity. Further research validating these signatures and exploring their mechanistic roles are warranted for the translation of these findings into clinical practice.

In summary, this study effectively used a metabolomics approach in a well-controlled intervention framework to unveil distinct metabolic signatures associated with high versus low weight loss responses in obese adolescents subjected to identical exercise and diet protocols. The high-weight-loss achievers demonstrated a metabolic phenotype characterized by enhanced fatty acid mobilization and oxidation, dynamic amino acid turnover, and indications of potentially more favorable gut microbiota–host interactions, all seemingly underpinned by a robust central energy pathway activity. These multifaceted metabolic distinctions advance beyond simply acknowledging the known heterogeneity in exercise-induced weight loss toward elucidating the potential intrinsic physiological determinants driving these differences when behavioral factors (such as compensatory eating) are minimized. However, despite the study’s design strictly controlling for exercise and relative energy intake, the observed metabolic differences may not be solely attributable to the magnitude of weight loss. Furthermore, a key limitation of this study is that the analysis focused exclusively on adolescents achieving the threshold for clinically significant weight loss (≥5%) and did not examine the metabolic profiles of those with <5% weight loss. Consequently, future research not only should consider including factors such as genetic background, baseline gut microbiota characteristics, and objective measurements of non-exercise activity thermogenesis to accurately disentangle intrinsic metabolic drivers from other potential confounders but should also specifically investigate metabolic changes in individuals with <5% weight loss to gain a comprehensive understanding of the spectrum of metabolic responses. Nevertheless, the differential metabolites and pathways identified in this study provide concrete biological evidence that explains the more effective response of some individuals subjected to the same energy deficit and exercise stimulus. These insights are valuable to the development of future personalized weight management strategies and prediction of weight loss responsiveness and guide the tailoring of interventions, such as nutritional support or specific exercise modalities, on the basis of individual metabolic profiles. These strategies may achieve more effective and sustainable outcomes and aid in addressing challenges in the fight against adolescent obesity. Further research is warranted to validate these metabolic signatures and explore their mechanistic roles, with the goal of translating these findings into clinical practice.

## 5. Conclusions

A four-week exercise intervention enhanced body morphology and physical fitness among obese adolescents, although it resulted in variable degrees of weight loss. Superior weight reduction outcomes were significantly correlated with the metabolites related to lipid, amino acid, organic acid, carbohydrate, and gut microbiota metabolism, and the enrichment of pathways involved in amino acid metabolism, fatty acid biosynthesis, and coenzyme A (CoA) biosynthesis. These findings indicate that intrinsic metabolic characteristics significantly influence individual responsiveness to exercise as a weight loss strategy.

## Figures and Tables

**Figure 1 metabolites-15-00313-f001:**
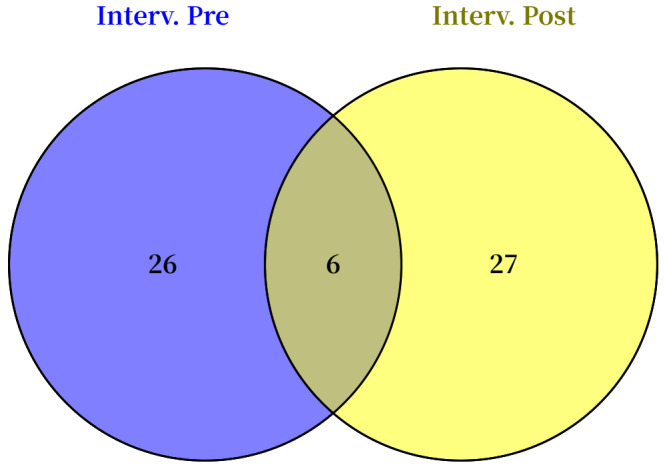
Venn diagram of different metabolites between HWL and LWL groups pre- and post-intervention.

**Figure 2 metabolites-15-00313-f002:**
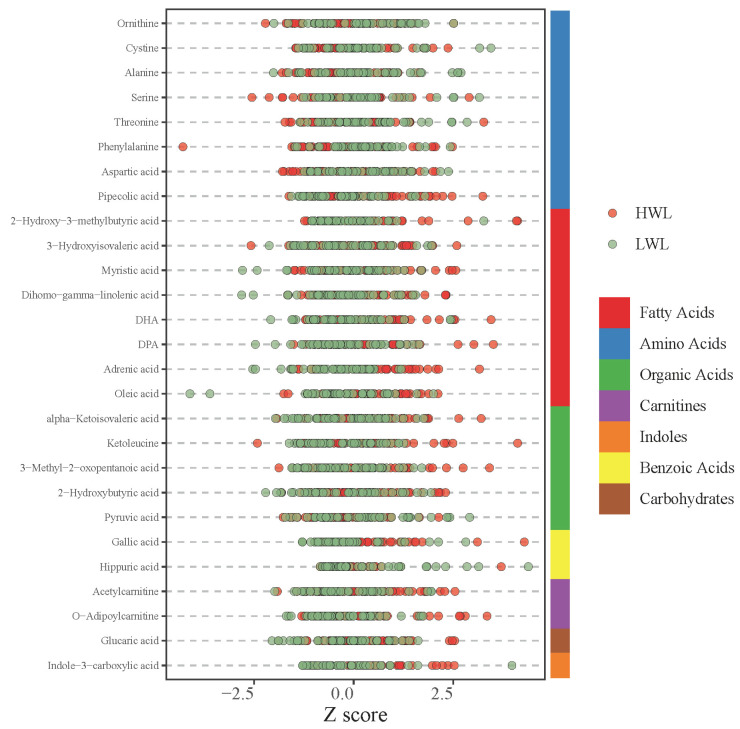
Z-score plot of different types of differential metabolites between LWL and HWL groups.

**Figure 3 metabolites-15-00313-f003:**
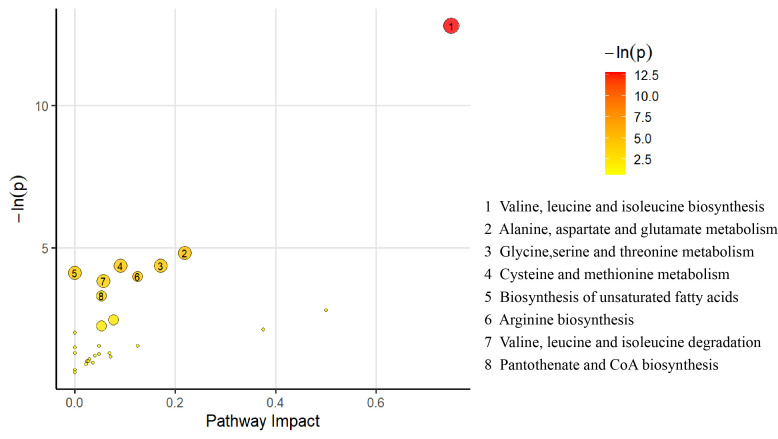
Pathway analysis bubble plot of differential metabolites.

**Table 1 metabolites-15-00313-t001:** Effects of exercise intervention on the body morphology and fitness level of obese adolescents.

Body Morphology and Fitness Level	Pre-Intervention (n = 98)	Post-Intervention (n = 98)	*p*	Cohen’s d	95%CI
Sex (male/female)	54/44	54/44	1.000	-	-
Age	13.05 ± 1.58	13.05 ± 1.69	1.000	-	-
Height (cm)	164.6 ± 9.8	164.6 ± 9.6	1.000	-	-
Weight (kg)	80.65 (71.35, 90.15)	72.35 (64.18, 81.23)	<0.001	0.55	8.31~9.7
Body mass index (kg/m^2^)	30.57 ± 4.41	27.26 ± 3.89	<0.001	0.79	3.07~3.55
Chest circumference (cm)	102.7 ± 9.5	95.0 ± 8.8	<0.001	0.83	6.74~8.57
Waist circumference (cm)	103.6 ± 11.8	94.7 ± 11.2	<0.001	0.77	8.06~9.77
Hip circumference (cm)	107.9 ± 10.1	100.7 ± 9.6	<0.001	0.73	6.60~7.87
Waist–hip ratio	0.96 ± 0.08	0.94 ± 0.09	<0.001	0.23	0.01~0.02
Waist–height ratio	0.63 ± 0.07	0.58 ± 0.06	<0.001	0.76	0.04~0.06
Body fat percentage (%)	32.68 ± 5.36	28.54 ± 6.04	<0.001	0.72	3.77~4.51
Body fat mass (kg)	27.73 ± 9.11	21.71 ± 8.02	<0.001	0.70	5.53~6.51
Free fat weight (kg)	55.64 ± 9.80	52.66 ± 9.08	<0.001	0.31	2.67~3.30
Body water (L)	40.06 ± 7.06	37.91 ± 6.53	<0.001	0.31	1.92~2.37
Skeletal muscle mass (kg)	50.92 ± 8.91	48.41 ± 8.30	<0.001	0.29	2.22~2.79

Note: - indicates not applicable for Cohen’s d or 95%CI calculation.

**Table 2 metabolites-15-00313-t002:** Comparison of body composition and fitness level between HWL and LWL groups before the exercise intervention.

Body Morphology and Fitness Level	LWL (n = 46)	HWL (n = 52)	*p*	Cohen’s d	95%CI
Sex (male/female)	19/27	35/17	0.009	-	-
Age	13.15 ± 1.59	12.96 ± 1.79	0.581	-	-
Height (cm)	163.0 (160.0, 171.7)	164.5 (159.0, 173.5)	0.933	-	-
Weight (kg)	78.45 (71.13, 86.25)	84.05 (72.58, 96.40)	0.180	0.27	−11.63~2.21
Body mass index (kg/m^2^)	28.95 (26.80, 31.70)	30.40 (28.28, 33.80)	0.108	0.32	−3.19~0.32
Chest circumferences (cm)	101.0 (94.1, 105.0)	102.0 (95.4, 109.3)	0.388	0.19	−5.85~2.29
Waist circumference (cm)	103.5 (94.6, 112.8)	105.8 (97.8, 112.0)	0.714	0.07	−5.63~3.87
Hip circumference (cm)	105.0 (100.0, 110.8)	110.3 (102.0, 115.5)	0.182	0.27	−6.77~1.30
Waist–hip ratio	0.98 (0.92, 1.01)	0.95 (0.91, 0.99)	0.292	−0.2	−0.01~0.04
Waist–height ratio	0.62 ± 0.07	0.63 ± 0.05	0.778	0.06	−0.03~0.02
Body fat percentage (%)	33.15 (29.23, 36.58)	32.70 (28.53, 35.98)	0.903	−0.02	−2.03~2.29
Body fat mass (kg)	37.80 (34.85, 42.00)	39.55 (36.10, 46.48)	0.501	0.13	−4.91~2.42
Free fat weight (kg)	25.75 (21.00, 30.68)	27.60 (21.53, 34.15)	0.081	0.36	−7.36~0.43
Body water (L)	52.50 (48.45, 58.35)	54.90 (50.15, 64.58)	0.081	0.36	−5.30~0.31
Skeletal muscle mass (kg)	48.15 (44.38, 53.75)	50.10 (45.86, 59.43)	0.077	0.36	−6.72~0.35

Note: - indicates not applicable for Cohen’s d or 95%CI calculation.

**Table 3 metabolites-15-00313-t003:** Comparison of changes in body morphology and fitness level between the HWL and LWL groups after the exercise intervention.

∆body Morphology and Fitness Level	LWL (n = 46)	HWL (n = 52)	aov_*p*	Cohen’s d	95%CI
∆weight (kg)	6.40 (5.30, 7.70)	10.45 (8.95, 12.85)	<0.001	1.80	−5.64~−3.55
∆body mass index (kg/m^2^)	2.45 (2.08, 2.80)	3.80 (3.30, 4.88)	<0.001	1.91	−1.98~−1.29
∆chest circumference (cm)	7.0 (5.0, 9.0)	8.0 (5.5, 11.0)	0.310	−0.20	−4.08~10.63
∆waist circumference (cm)	6.7 ± 1.9	10.2 ± 4.6	<0.001	0.80	−4.76~−1.56
∆hip circumference (cm)	4.5 (2.5, 7.7)	5.0 (2.5, 8.3)	0.029	0.52	−2.60~−0.27
∆waist–hip ratio	0.01 ± 0.02	0.03 ± 0.04	0.083	0.34	−0.02~0.001
∆waist–height ratio	0.04 (0.03, 0.05)	0.06 (0.04, 0.09)	<0.001	0.76	−0.02~−0.008
∆body fat percentage (%)	3.35 ± 1.30	4.63 ± 2.00	<0.001	0.73	−2.50~−1.21
∆body fat mass (kg)	4.63 ± 1.11	7.15 ± 2.03	<0.001	1.49	−3.68~−2.10
∆Free fat weight (kg)	1.95 (1.50, 2.50)	3.65 (2.80, 4.38)	<0.001	1.31	−2.23~−1.17
∆body water (L)	1.40 (1.10, 1.80)	2.65 (2.00, 3.18)	<0.001	1.31	−1.61~−0.84
∆skeletal muscle mass (kg)	1.60 (1.18, 2.10)	3.10 (2.30, 3.80)	<0.001	1.21	−1.94~−0.96

Note: ∆ indicates the amount of change in the indicator (pre-intervention − post-intervention); “aov_*p*” represents the *p*-value adjusted for sex.

**Table 4 metabolites-15-00313-t004:** Between-group changes in differential metabolites before intervention.

Metabolite	Class	LWL	HWL	aov_*p*
Glutamic acid	Amino Acids	41.44 (34.14, 48.11)	47.48 (35.95, 63.06)	0.008
beta-Alanine	Amino Acids	4.06 ± 1.06	4.57 ± 1.14	0.024
Dimethylglycine	Amino Acids	5.03 (4, 5.83)	5.74 (4.51, 6.86)	0.005
Hydroxyphenyllactic acid	Phenylpropanoic Acids	1.56 (1.34, 1.78)	1.70 (1.58, 2.14)	0.006
Indole-3-methyl acetate	Indoles	0.12 (0.08, 0.21)	0.09 (0.06, 0.14)	0.019
Indoleacetic acid	Indoles	2.12 (1.63, 2.71)	1.94 (1.40, 2.25)	0.023
Cinnamic acid	Phenylpropanoids	0.07 (0.06, 0.08)	0.06 (0.06, 0.07)	0.001
2-Phenylpropionate	Phenylpropanoic Acids	0.27 (0.22, 0.45)	0.22 (0.12, 0.31)	0.018
Hydrocinnamic acid	Phenylpropanoic Acids	0.28 (0.21, 0.43)	0.22 (0.17, 0.31)	0.031
Benzenebutanoic acid	Benzenoids	0.1 (0.09, 0.11)	0.10 (0.09, 0.10)	0.037
Glyceric acid	Carbohydrates	3.91 (3.11, 4.25)	4.39 (3.76, 4.86)	0.003
N-Acetylserine	Amino Acids	1.66 ± 0.29	1.77 ± 0.27	0.049
N-Acetylneuraminic acid	Carbohydrates	0.71 (0.67, 0.80)	0.78 (0.69, 0.88)	0.017
Pyroglutamic acid	Amino Acids	44.9 (36.23, 51.71)	51.03 (38.19, 71.02)	0.007
Maltotriose	Carbohydrates	3.48 (1.54, 4.46)	1.87 (1.14, 3.44)	0.045
Rhamnose	Carbohydrates	0.93 (0.75, 1.13)	1.25 (0.88, 1.42)	0.010
Propionic acid	SCFAs	2.41 (1.90, 3.46)	3.15 (1.99, 5.63)	0.028
Homovanillic acid	Phenols	1.14 (1.03, 1.31)	1.25 (1.09, 1.48)	0.006
Butyric acid	SCFAs	2.21 ± 0.50	2.47 ± 0.50	0.011
Phenylacetylglutamine	Amino Acids	1.21 (0.78, 2.02)	0.92 (0.49, 1.37)	0.020
Valeric acid	SCFAs	0.76 (0.58, 1.17)	0.79 (0.52, 2.66)	0.021
2-Methylpentanoic acid	SCFAs	0.26 (0.15, 0.41)	0.21 (0.13, 1.22)	0.009
Heptanoic acid	Fatty Acids	0.44 ± 0.17	0.54 ± 0.22	0.016
TCDCA	Bile Acids	0.06 (0.03, 0.11)	0.1 (0.04, 0.17)	0.034
Oxoglutaric acid	Organic Acids	65.84 (58.41, 76.69)	76.58 (65.22, 101.62)	0.000
Oxoadipic acid	Organic Acids	0.13 (0.11, 0.15)	0.14 (0.12, 0.17)	0.026
Undecylenic acid	Fatty Acids	0.87 (0.58, 1.29)	0.74 (0.45, 1.10)	0.025
Arachidonic acid	Fatty Acids	90.04 ± 28.22	108.87 ± 28.60	0.001
DPAn-6	Fatty Acids	2.00 (1.63, 2.50)	2.42 (2.09, 2.82)	0.008
Heptadecanoic acid	Fatty Acids	11.41 (7.62, 14.24)	9.2 (6.79, 11.72)	0.027
Succinic acid	Organic Acids	2.78 ± 0.46	3.00 ± 0.45	0.017
Methylmalonylcarnitine	Carnitines	0.04 (0.04, 0.04)	0.04 (0.04, 0.05)	0.007

Note: “aov_*p*” represents the *p*-value adjusted for sex.

**Table 5 metabolites-15-00313-t005:** Between-group changes in differential metabolites after intervention.

Metabolite	Class	LWL	HWL	aov_*p*
Alanine	Amino Acids	305.71 ± 64.39	277.43 ± 45.39	0.011
Aspartic acid	Amino Acids	1.91 (1.47, 2.38)	1.54 (1.10, 2.23)	0.028
Cystine	Amino Acids	82.03 (69.04, 96.33)	64.88 (47.02, 87.40)	0.042
Serine	Amino Acids	176.4 (154.18, 187.24)	161.88 (152.26, 181.80)	0.047
Ornithine	Amino Acids	26.61 ± 5.08	24.56 ± 4.91	0.041
Phenylalanine	Amino Acids	51.24 (45.22, 55.64)	45.55 (38.98, 51.16)	0.031
Pipecolic acid	Amino Acids	7.41 (6.09, 8.43)	7.68 (6.76, 9.89)	0.015
Threonine	Amino Acids	61.3 (55.48, 69.54)	52.29 (47.91, 64.84)	0.001
Gallic acid	Benzoic Acids	0.53 (0.34, 0.78)	0.81 (0.54, 1.22)	0.043
Glucaric acid	Carbohydrates	0.07 ± 0.02	0.09 ± 0.02	0.002
2-Hydroxy-3-methylbutyric acid	Fatty Acids	22.59 (15.72, 31.04)	27.22 (19.78, 38.25)	0.035
Hippuric acid	Benzoic Acids	0.71 (0.28, 2.06)	0.47 (0.29, 0.95)	0.014
Maltotriose	Carbohydrates	1.67 (0.73, 3.13)	1.05 (0.61, 1.76)	0.008
Acetylcarnitine	Carnitines	73.54 (65.22, 96.24)	86.9 (70.82, 110.28)	0.034
O-Adipoylcarnitine	Carnitines	0.11 (0.1, 0.14)	0.14 (0.11, 0.15)	0.021
3-Hydroxyisovaleric acid	Fatty Acids	1.13 ± 0.33	1.30 ± 0.36	0.011
3-Methyl-2-oxopentanoic acid	Organic Acids	108.55 ± 22.36	118.9 ± 28.61	0.043
Adrenic acid	Fatty Acids	7.60 ± 2.13	9.47 ± 2.73	0.000
Arachidonic acid	Fatty Acids	106.42 ± 28.97	131.34 ± 30.65	0.000
DHA	Fatty Acids	44.89 (37.01, 59.12)	53.52 (45.96, 74.57)	0.009
Dihomo-gamma-linolenic acid	Fatty Acids	5.89 ± 1.90	7.00 ± 2.13	0.005
Pyruvic acid	Organic Acids	129.31 (86.48, 178.64)	108.63 (84.75, 129.40)	0.029
DPA	Fatty Acids	4.36 (3.61, 5.60)	5.13 (4.12, 6.73)	0.010
DPAn-6	Fatty Acids	2.41 (1.87, 2.86)	2.97 (2.34, 3.87)	0.000
Myristic acid	Fatty Acids	33.40 ± 9.44	37.70 ± 10.96	0.040
Undecylenic acid	Fatty Acids	0.75 (0.49, 1.11)	0.56 (0.39, 0.87)	0.018
Indole-3-carboxylic acid	Indoles	0.10 (0.08, 0.13)	0.13 (0.09, 0.16)	0.045
2-Hydroxybutyric acid	Organic Acids	187.38 ± 70.44	214.57 ± 64.22	0.025
alpha-Ketoisovaleric acid	Organic Acids	28.39 ± 5.00	30.73 ± 5.53	0.023
Ketoleucine	Organic Acids	292 (240, 327.52)	306.47 (275.62, 342.72)	0.015
Oleic acid	Fatty Acids	923.92 (841.54, 1060.98)	1007.44 (891.57, 1107.10)	0.029
Homovanillic acid	Phenols	1.19 (1.01, 1.31)	1.32 (1.13, 1.45)	0.030
Hydroxyphenyllactic acid	Phenylpropanoic Acids	1.63 (1.41, 1.96)	1.89 (1.54, 2.06)	0.015

Note: “aov_*p*” represents the *p*-value adjusted for sex.

**Table 6 metabolites-15-00313-t006:** ROC curve of differential metabolites.

Metabolite	AUC	CI1	CI2	Thres	Specificity	Sensitivity
Threonine	0.704	0.599921	0.80727	54.840	0.577	0.826
Adrenic acid	0.691	0.585515	0.797428	9.185	0.596	0.783
Glucaric acid	0.659	0.550605	0.766703	0.087	0.462	0.783
Phenylalanine	0.659	0.549571	0.768991	48.860	0.654	0.652
3-Hydroxyisovaleric acid	0.658	0.548517	0.767118	1.077	0.808	0.522
DHA	0.648	0.538772	0.757214	45.804	0.750	0.522
Gallic acid	0.642	0.530811	0.75347	0.543	0.750	0.543
Cystine	0.637	0.524336	0.749076	67.288	0.538	0.804
Aspartic acid	0.632	0.520945	0.742851	1.389	0.462	0.826
Indole-3-carboxylic acid	0.631	0.520346	0.742614	0.131	0.538	0.739
Pipecolic acid	0.629	0.518154	0.739789	6.441	0.846	0.391
Dihomo-gamma-linolenic acid	0.628	0.517094	0.738759	7.064	0.500	0.804
DPA	0.628	0.517179	0.73951	4.894	0.577	0.674
Alanine	0.627	0.514406	0.738938	334.750	0.904	0.326
Acetylcarnitine	0.626	0.513978	0.737694	77.560	0.673	0.587
Ketoleucine	0.621	0.509903	0.732572	260.508	0.827	0.413
O-Adipoylcarnitine	0.62	0.507909	0.731639	0.109	0.750	0.500
Ornithine	0.62	0.506404	0.732727	25.511	0.635	0.674
Oleic acid	0.612	0.499796	0.724284	948.005	0.673	0.543
2-Hydroxy-3-methylbutyric acid	0.610	0.497617	0.722283	26.585	0.538	0.674
alpha-Ketoisovaleric acid	0.607	0.4938	0.719411	24.856	0.923	0.283
2-Hydroxybutyric acid	0.606	0.491841	0.719697	137.898	0.942	0.304
3-Methyl-2-oxopentanoic acid	0.599	0.48667	0.712326	105.097	0.673	0.500
Pyruvic acid	0.598	0.480857	0.714795	127.578	0.750	0.522
Serine	0.594	0.481091	0.707873	165.999	0.538	0.674
Hippuric acid	0.593	0.477211	0.708826	1.895	0.962	0.283
Myristic acid	0.591	0.477502	0.703937	40.597	0.404	0.804

## Data Availability

The datasets used and/or analysed during the current study are available from corresponding authors on reasonable request.

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
