# Peer review of "Metabolic Characteristics of Obese Adolescents with Different Degrees of Weight Loss After Identical Exercise Training Intervention"

_metabolites, 2025, doi:10.3390/metabo15050313_

Round 1
Reviewer 1 Report
Comments and Suggestions for Authors
Line 20: We can use “exercise training intervention” instead of “exercise intervention”.
Line 37: enhance body morphology change (is more clear) than just “enhance body morphology”
Line 48: Specify the time period for the WHO statistics.
Lines 52-53: Provide more references for obesity-related complications (especially from ACSM references)
Lines 63-64: The statement about "individual differences in weight loss effects" needs supporting references
The mechanism linking exercise to metabolic changes should be more thoroughly explained.
(Lines 68-73) The description of metabolomics technology requires expansion more detail specifically in trained and untrained people. (Not just obese people)
Lines 74-80: The aims could be more specific (probably mechanisms) about expected outcomes.
What is the indicator of reaching adolescent for the subjects of this research?
Demographic characteristics: It is suggested that demographic characteristics (age, sex, initial body mass index, fitness level) be presented.
Grouping criteria: It is not clear why the 10% weight loss threshold was considered for the HWL group. References to reputable studies could strengthen this section. Duration of 4 weeks for such loosing is consider more than ACSM guidelines and we need reference for this amount of lose weight.
It is better to report total lose weight by mean±SD for all 103 subjects.
It is better that we have distribution of lose weight amount among this 103 subjects (eg: how much is range for range 68% of subjects).
Number of sample is 103, why? Did we have any loos or miss in subjects during this 4-week? How many people do we have in each group?
LINE 111: What is the meaning of “informal training”? What does mean? It is better Ex RX Guide Lines be clear. (FIIT)
How much is calorie expenditure per week for this intervention? How much is negative calorie per week? Why is difference lose weight?
In case of dietary control, monitoring dietary adherence: It is unclear whether food diary, 24-hour recall interview, or nutritional monitoring software was used to assess adherence to the diet.
Effects of exercise training Intervention on body composition and Glucose-Lipid Metabolism. Table 1 reports a significant decrease in Skeletal Muscle Mass (SMM). This is metabolically important, as decreased muscle mass can reduce basal metabolism and affect the long-term effects of weight loss. Line 416: In discussion part, it is better to discuss about reducing of SMM and probably its side effects.
It is unclear whether age, sex, initial fitness level, or initial body composition influenced the rate of weight loss. It is suggested that a covariance or multivariate regression model be used to control for the effects of some of these variables.
It is unclear whether the metabolomics profile differs between groups solely due to the amount of weight loss or whether other factors (e.g., differences in diet composition or compliance with the exercise intervention) may also play a role. At least new questions should be present in this part.
The analysis of metabolic pathways is incomplete: Table 5 only provides a list of differential metabolites but does not explain the physiological role of these metabolites in weight loss or the differences between HWL and LWL.

Reviewer 2 Report
Comments and Suggestions for Authors
- The paper is well-written, with a clear structure and logical flow. The use of metabolomics to study weight loss in obese adolescents is innovative and provides valuable insights.
- Suggestions: Addressing the suggestions mentioned above would enhance the clarity and impact of the paper. Additionally, proofreading for minor grammatical errors and ensuring consistency in formatting would improve the overall quality.
Flow and Readability: The paper flows well, but simplifying complex sentences and ensuring clarity can enhance readability
Reviewer 3 Report
Comments and Suggestions for Authors
Thank you for the opportunity to revise the current MS. Please find my comments and suggestion below, line by line:
Abstract:
Line 24-26: Ok, but, by how much ?
Intro: In general good, but too short for my taste. I would only ask authors to add couple sentences to build study rationale and study hypotheses.
Methods: Line 130 - 133, perhaps redundant, these are all well known facts.
Line 135-139: Body positions during measurement (upright vs supine), Electrode frequency during data collection (in Hz).
Line 226: Please consider adding effect sizes and 95% CI.
Table 1: Height should not be given in two decimal places, or you were able to collect data in mm using some novel approach ? Same goes for Table 3.
Line 407: Could you expand on that please ?
The discussion seems a bit one sided, and it is not uncommon that people how start to exercise, start eating more. Could you comment on this please and see the work of prof Hall and others
On the pathogenesis of obesity: causal models and missing pieces of the puzzle | Nature Metabolism
Gut‐derived appetite hormones do not explain energy intake differences in humans following low‐carbohydrate versus low‐fat diets - Hengist - 2024 - Obesity - Wiley Online Library
Models of body weight and fatness regulation | Philosophical Transactions of the Royal Society B: Biological Sciences
Reviewer 4 Report
Comments and Suggestions for Authors
The article titled "A study of Metabolic Characteristics of Obese Adolescents with Different Degrees of Weight Loss after Identical Exercise Intervention” highlights the difference between Low Vs High weight loss groups by comparing metabolic profiles.
Major comments:
1. While other clinical parameters have been summarized, it is not clear if the study groups consisted of both male and female participants. There are reports indicating differences in metabolism and metabolic profiles between sexes and it is unclear how this study accounts for such factors. Also, details on age distribution within and between groups are lacking.
2. The study investigates individuals with weight loss between 5-10% and >10% and compares their metabolic profiles to draw an association with weight loss. However, this approach cannot ascertain whether differences in metabolic changes are the underlying cause of weight loss. As:
a. The study does not include basal metabolic profiles to support this conclusion. So, diagnostic efficiency (section 3.3.4) can’t be determined. Did participants in the two groups have comparable metabolic profiles before exercise intervention? From another perspective, if study participants at the beginning of the study already had varying profiles and were not measured, the latter measurement would make it look like exercise-induced changes. Although authors report clinical characteristics from LWL and HWL groups in a retrospective approach, it does not truly indicate their metabolic state.
b. The measurement only includes upwards of 5% loss. The changes in metabolic profiles on <5 % weight loss have not been estimated. Would any of the metabolic differences be present even if there is <5 % weight loss?
c. Short-term exercise has been reported to reduce blood volume. Would this affect the metabolic profiles?
Minor comments:
1. Line 58-66: References required.
2. Section 2.4.1: The abbreviation for microliter needs to be corrected.
3. Line 160: Reference to section 2.2.3 should be corrected as there is no such section.
4. Line 230: Rephrasing is necessary.
